# Efficacy and Safety of 6-Month High Dietary Protein Intake in Hospitalized Adults Aged 75 or Older at Nutritional Risk: An Exploratory, Randomized, Controlled Study

**DOI:** 10.3390/nu15092024

**Published:** 2023-04-22

**Authors:** Shota Moyama, Yuichiro Yamada, Noboru Makabe, Hiroki Fujita, Atsushi Araki, Atsushi Suzuki, Yusuke Seino, Kenichiro Shide, Kyoko Kimura, Kenta Murotani, Hiroto Honda, Mariko Kobayashi, Satoshi Fujita, Koichiro Yasuda, Akira Kuroe, Katsushi Tsukiyama, Yutaka Seino, Daisuke Yabe

**Affiliations:** 1Yutaka Seino Distinguished Center for Diabetes Research, Kansai Electric Power Medical Research Institute, Kyoto 604-8436, Japan; 2Center for Metabolism and Clinical Nutrition, Kansai Electric Power Hospital, Osaka 553-0003, Japan; 3Center for Diabetes, Endocrinology and Metabolism, Kansai Electric Power Hospital, Osaka 553-0003, Japan; 4Department of Human Life Sciences, Mimasaka University, Tsuyama 708-8511, Japan; 5Department of Endocrinology and Diabetes and Geriatric Medicine, Akita University School of Medicine, Akita 010-0852, Japan; 6Department of Diabetes, Metabolism, and Endocrinology, Tokyo Metropolitan Geriatric Hospital, Tokyo 173-0015, Japan; 7Department of Endocrinology and Metabolism, Fujita Health University, Toyoake 470-1192, Japan; 8Department of Metabolism and Clinical Nutrition, Kyoto University Hospital, Kyoto 606-8397, Japan; 9Department of Nutritional Sciences, Morioka University, Takizawa 020-0605, Japan; 10Biostatistics Center, Graduate School of Medicine, Kurume University, Kurume 830-0011, Japan; 11Department of Physical Therapy, Faculty of Health Sciences, Aino University, Ibaraki 567-0012, Japan; 12Department of Health and Welfare, Settsu City Hall, Settsu 566-8555, Japan; 13College of Sport and Health Sciences, Ritsumeikan University, Kusatsu 525-8577, Japan; 14Department of Diabetes and Endocrinology, Saiseikai-Noe Hospital, Osaka 536-0001, Japan; 15Department of Diabetes and Metabolism, Hikone Municipal Hospital, Hikone 522-0057, Japan; 16Center for Diabetes, Kobayashi Memorial Hospital, Hekinan 447-0863, Japan; 17Department of Diabetes, Endocrinology and Metabolism, Gifu University Graduate School of Medicine, 1-1 Yanagido, Gifu 501-1194, Japan; 18Department of Rheumatology and Clinical Immunology, Gifu University Graduate School of Medicine, Gifu 501-1194, Japan; 19Center for One Medicine Innovative Translational Research, Gifu University, Gifu 501-1194, Japan; 20Center for Preemptive Food Research, Gifu University Institute for Advanced Study, Gifu 501-1194, Japan; 21Center for Research, Education and Life Design, Gifu University, Gifu 501-1194, Japan

**Keywords:** frailty, malnutrition, older adults, protein intake, dietary intervention

## Abstract

The aim of this study was to investigate the effects of increased dietary protein in daily-life settings in Japan for 6 months on the activities of daily living (ADL) in adults aged 75 or older at nutritional risk. The study was an open-label, exploratory, randomized controlled trial conducted at seven hospitals in Japan. The study participants were adults aged 75 or older who were hospitalized for treatable cancer, pneumonia, fractures, and/or urinary-tract infection at nutritional risk. The primary outcome was change in grip strength, skeletal muscle, and ADL indices (Barthel index, Lawton score). One hundred sixty-nine patients were randomly assigned to the intensive care (IC) or standard care (SC) group; the protein intake goals (g/kgw/day) were 1.5 for IC and 1.0 for SC. There was a significant improvement in grip strength only in the IC group (1.1 kg: 95% CI 0.1 to 2.1) (*p* = 0.02). While the skeletal muscle index and ADL indices were not significantly improved in either group, the improvement ratio tended to be greater in the IC group. There was no decrease in renal function in either group. Thus, intervention of increased dietary protein in daily-life settings for 6 months in adults aged 75 or older with treatable cancer, pneumonia, fractures, and/or urinary-tract infection and at nutritional risk may be effective in ameliorating loss of muscle strength.

## 1. Introduction

Increased life expectancy is an important feature of developed societies, highlighting the need for a healthy life span that enables independent living for older adults without the need of specialized medical care. Frailty, which is an intermediate stage between independence and requirement of nursing care, is a condition that can be reversed to facilitate recovery from acute illness, trauma, and inflammation. In older adults having various chronic diseases, nutritional improvement might well represent an opportunity to moderate frailty and empower better activities of daily living (ADL).

Indeed, it has been reported that older adults may have insufficient protein intake for health maintenance [1]. The ability to synthesize muscle protein deteriorates with age, resulting in a need for higher protein intake by older adults than that by younger adults [2]. In fact, elderly adults are reported to require 1.2 to 1.5 g protein per kg body weight per day (g/kgw/day) to maintain health and as much as 2.0 g/kgw/day in cases of severe disease and/or malnutrition [3]. Previous studies suggested that 12-week protein enrichment was effective in improving physical function in older adults [4,5,6], but medium to long term effects remained unclear due to the relatively short observation periods. While some studies have examined the effect of intervention for 6 months [7], evidence on medium to long term effects of increased dietary protein intake in older adults is sparse [8]. There are also concerns regarding the protein burden on renal function, which makes careful monitoring of dietary protein supplementation in older adults necessary [9,10,11].

This randomized, comparative clinical trial was conducted to clarify the safety and efficacy of increased dietary protein intake in daily-life settings for 6 months in adults aged 75 or older.

## 2. Materials and Methods

### 2.1. Study Design

A multi-center, open-label, two-arm, randomized, comparative clinical trial was performed to evaluate efficacy and safety of increased dietary protein for 6 months for maintaining physical function and ADL in adults aged 75 or older with nutritional risk. The study was approved by the Japan Society of Metabolism and Clinical Nutrition Research Ethics Committee (approval no.: JSMCN 18-001-01) and the Study Protocol was made publicly available on the UMIN Clinical Trials Registry (Clinical trial registration number UMIN000032813 and date of registration 31 May 2018). The study was performed in accordance with the guidelines in the Declaration of Helsinki and was prepared in accordance with CONSORT standards. All participants were given written and oral explanations and informed consent was obtained.

### 2.2. Participants

All participants were adults aged 75 or older who were admitted to one of 7 participating hospitals in Japan for treatable cancer, pneumonia, fractures, and/or urinary-tract infection and were at nutritional risk, scoring below 11 on the Mini Nutritional Assessment Short Form, a simple evaluation questionnaire for nutritional state [12,13]. Participants who met one or more of the following criteria were excluded:(a)Dementia (Mini-Cog score below 3 [14]) or psychiatric disorder.(b)Current steroid use.(c)Moderate or severe liver disease (Child-Pugh classification above 7) or moderate to severe renal dysfunction (serum creatinine above 2.0 mg/dL).(d)Heart failure that restricted physical activity (New York Heart Association functional classification of II or greater).(e)Severe, chronic complications of diabetes such as overt proteinuria, proliferative retinopathy or severe neuropathy.(f)Infectious disease of the systemic inflammatory reaction group or trauma (4 or higher on the abbreviated injury scale).(g)Gastrointestinal reconstruction surgery.(h)Untreated cancer.(i)Cardiac pacemaker or implanted defibrillator.(j)Limb motor paralysis due to central nervous system disease.(k)Participation in this clinical study deemed not appropriate.

### 2.3. Target Sample Size and Rationale

As this was an exploratory randomized controlled trial, 200 participants were set as the target sample size to be recruited through the participating facilities during the study period. One hundred and sixty-nine patients were finally enrolled and assigned to the study.

### 2.4. Randomization and Masking

Participants were randomly allocated in a 1:1 ratio to the intensive care (IC) group or the standard care (SC) group by the randomly permuted block method with stratification by gender, skeletal mass index (SMI), and primary disease of hospitalization (cancer, fractures, urinary-tract infection, or pneumonia). A computer-generated random number sequence was prepared by an independent statistician before initiation of the study; the sequence was managed on a password-protected computer by an external data manager having no connection with the survey. After recruitment, the participants were numbered consecutively and the information was sent to the external data manager, who finally sent the group allocations back to the survey personnel. After allocation to the groups, it was not feasible for the participants or study personnel to be blinded; all personnel who subsequently evaluated the data or made judgment on the results were blinded.

### 2.5. Procedures

Each participant was examined and treated by a physician and given individualized nutritional advice by a registered dietician before discharge from the hospital (baseline) and 3 and 6 months after discharge. Treatment by the physician involved only standard care for the primary disease and concomitant conditions. The nutritional interventions for the two groups were as follows: IC, energy 25–35 kcal/kgw/day, protein 1.5 g/kgw/day and vitamin D 10 μg/day; SC, energy 25–35 kcal/kgw/day and protein 1.0 g/kgw/day. Individualized intervention by a registered dietician was conducted based on the results of an interview and the Food Frequency Questionnaire Based on Food Groups (FFQg) [15]. The intervention advice was given not to simply increase total daily protein intake, but rather to maintain an appropriate protein intake rate over the three meals of the day. Neither group received individualized advice on exercise; both groups received a leaflet asking the participants to perform light resistance exercise 6 times a week (3 exercises, each one repeated 10 times but not on two consecutive days) for the arms and legs (side lateral raise and thigh lift exercise). All outcomes were assessed at baseline and 3- and 6 months after intervention and included blood tests (renal and hepatic function). Arm strength was measured using a Smedley-type grip-strength meter (TKK 5401; Takei Scientific Instruments Co., Ltd., Niigata, Japan); leg strength [ground reaction force when standing up from a chair (maximal rate of force development (Δ87.5 ms) per unit body weight (RFD8.75/w)) and speed standing up from a chair (peak reaction force per unit body weight (F/w))] was measured using a specialized body weight scale (BM-220; Tanita Corporation, Tokyo, Japan) [16]. To measure grip strength, the subject held a hand dynamometer downward in the standing position; grip strength was measured twice on each side at maximum effort while exhaling and the maximum value was used as upper limb muscle strength. ADL was evaluated on the bases of the Barthel index [17] and Lawton score [18]. SMI was evaluated by bioelectrical impedance analysis using a multi-frequency body composition meter at seated position after 5 min resting in accordance with the general instructions by the manufacturer (Model JMW140, InBody S10; Biospace Co., Ltd., Seoul, Republic of Korea) [19]. The cutoff point of grip strength and SMI were determined by the diagnostic criteria of the Asian Working Group for Sarcopenia (AWGS2019) (SMI: <7.0 kg/m^2^ for male and <5.7 kg/m^2^ for female; grip strength: <28.0 kg for male and <18.0 kg for female) [20].

Intake of energy and macronutrients was evaluated using the FFQg, which includes questions on 29 food groups and 10 preparation methods for each food group and allows daily intake frequency and total intake per week to be quantified [15]. For evaluation of physical activity, total energy consumption due to physical activity per day was calculated using the International Physical Activity Questionnaire (IPAQ), which includes questions regarding the number of days per week on which the subject performed moderate to high intensity physical activities including leisure, housework, employment, physical activity on the move and the total duration of these activities per week [21].

### 2.6. Outcomes

The primary endpoint was change in grip strength, SMI and ADL indices over 6 months; a randomized controlled trial in older adults has shown that protein-enhancing intervention can improve grip strength [22,23]. The secondary endpoint was change in macronutrient intake over 6 months. In addition, changes in the following parameters over 6 months were evaluated: leg muscular strength (RFD8.75/w and F/w), body mass index (BMI), hemoglobin (Hb), creatinine (Cre), blood urea nitrogen (BUN), estimated glomerular filtration rate (eGFR), albumin and C-reactive protein (CRP). Blood tests were performed at each facility in the early morning after fasting for at least 12 h.

### 2.7. Statistical Analysis

After excluding participants who dropped out or had data missing, the number included in the per-protocol analysis was 93, 44 in the IC group and 49 in the SC group. The continuous variables were collated as the number of measurements, median and interquartile range (IQR) or mean and 95% confidence interval (CI); *t*-test or Mann–Whitney U test was used to evaluate differences between the groups at baseline. The Kolmogorov–Smirnov normality test was used to test normality of the data. Categorical variables are shown as frequency and percentage; differences between groups at baseline were evaluated using Fisher’s exact test. Using a two-way, repeated analysis-of-variance model, the differences between groups up to 6 months after baseline and their interactions with inter- and intra-subject factors were verified. For this purpose, the group and time point were included in the model and the interactions were calculated for each variable. Within each group, one-way, repeated analysis of variance and Bonferroni-corrected multiple comparison test were used to assess the statistical significance of changes up to 6 months after baseline. In all statistical tests, the significance level was set at 5% on both sides; *p*-values below 0.05 were considered statistically significant. Statistical Package for Social Sciences, ver. 26 (SPSS, Inc., Chicago, IL, USA) was used for data analysis.

## 3. Results

### 3.1. Characteristics of the Participants

Of the 201 participants recruited, 32 were excluded after assessment for eligibility. Of the remaining 169, 83 were allocated to the IC group and 86 to the SC group (Figure 1). Those who dropped out or had missing data were excluded from the analysis, leaving a total of 93, 44 in the IC group and 49 in the SC group. A considerable number of participants were excluded from the analysis because they requested a transfer to a general practitioner near their home. Table 1 shows participant demographic characteristics, blood test results, nutrient intake, physical activity, primary disease for hospitalization, and concomitant diseases at baseline. The gender ratios were consistent in the two groups; there were no significant differences in age, BMI, limb muscular strength, SMI, ADL level, blood test results, or physical activity. Intake of energy, protein and carbohydrate at baseline were lower in the IC group than those in the SC group: energy (kcal/kgw/day), IC 24.4 (22.5–28.4) and SC 29.1 (24.7–33.2) (*p* < 0.05); protein (g/kgw/day), IC 1.0 (0.9–1.2) and SC 1.2 (1.0–1.4) (*p* < 0.05); and carbohydrate (g/day), IC 211.7 (176.1–236.8) and SC 222.6 (195.3–245.1) (*p* < 0.05). The ADL-related indices (Barthel index and Lawton score) at baseline were similar in both groups. Hb and albumin were relatively low in both groups. Cancer was the primary disease for hospitalization in ≥50% of the participants, followed by pneumonia. Non-communicable diseases such as hypertension, dyslipidemia, and diabetes were major concomitant diseases in both groups.

### 3.2. Muscle Strength, Skeletal Muscle Mass, Physical Function, and Blood Examination

Grip strength was increased more after 6 months in the IC group than that in the SC group, with significant inter-group difference (Figure 2). SMI and RFD8.75/w but not F/w were significantly increased after 6 months in both groups; the increases tended to be greater in the IC group but did not reach statistical significance (Table 2). BMI was significantly increased after 6 months but only in the IC group. No differences were found in the Barthel index or Lawton score before or after 6 months in either group. Albumin was increased and CRP was decreased significantly in both groups after 6 months (*p* < 0.01). eGFR and Cre were similar before and after 6 months in both groups.

### 3.3. Nutrient Intake

Energy intake [kcal/kgw/day: IC, baseline 24.4 (22.5–28.4) and 6 month 32.1 (28.1–37.1) (*p* < 0.01); SC, baseline 29.1 (24.7–33.2) and 6 month 32.1 (27.5–40.3) (*p* < 0.01)]; fat intake [g/day; IC, baseline 40.7 (33.2–45.6) and 6 month 62.9 (48.2–69.6) (*p* < 0.01); SC, baseline 42.4 (38.2–49.1) and 6 month 55.3 (44.7–66.0) (*p* < 0.01)], all of which tended to increase after 6 months in both groups, but a statistically significant increase was found only in the IC group for protein intake [g/kgw/day; IC, baseline 1.0 (0.9–1.2) and 6 month 1.3 (1.1–1.5) (*p* < 0.01); SC, baseline 1.2 (1.0–1.4) and 6 month 1.2 (1.0–1.5)] and carbohydrate intake [g/day; IC, baseline 211.7 (176.1–236.8) and 6 month 249.1 (216.4–289.8) (*p* < 0.01); SC, baseline 222.6 (195.3–245.1) and 6 month 244.6 (204.5–275.6)]. No significant changes in vitamin D intake or physical activity were noted in either group. Table 3 shows the changes in intake of the various food groups that are protein sources, showing that intake of soy products, meat and eggs were increased significantly only in the IC group.

## 4. Discussion

The present study is the first multicenter, randomized, comparative clinical trial to examine the efficacy and safety of increased dietary protein intake for improving physical activity in older adults with diseases such as cancer, pneumonia, fractures, and urinary tract infections and at nutritional risk.

Although intervention studies with supplements such as leucine and whey protein have been shown to be effective in improving ADL in older adults [24,25,26], the efficacy and safety of dietary interventions that do not employ supplements have not been evaluated by randomized clinical trial. In addition, there are few studies on nutritional intervention in patients with temporarily decreased ADL due to cancer, infection, or trauma. Such patients often have insufficient dietary energy and protein intake during hospitalization and afterward, which exacerbates physical decline [27,28]. In fact, it has been reported that one week of hospitalization can decrease muscle mass by as much as 3.2 ± 0.9% per day [29], about twice the usual decrease with aging [30].

In the present study, we found that grip strength was increased significantly, and that SMI and leg strength tended to be improved by 6-month nutritional intervention of increased dietary protein intake, while there was no change in the ADL-related indices. According to IPAQ quantification, the physical activity recommended to both groups was light-intensity activity such as walking 3 to 5 times per week; since physical activity of the two groups was similar, exercise should not be a factor in the increase of muscle strength and mass observed in the IC group. It has been reported that grip strength decreases by 0.4 kg (0.5 kg for men, 0.3 kg for women) per year with aging [31]; even so, while neither group showed a decline in grip strength, the intervention group showed improvement. The reason why the ADL index was not seen to improve may be partly due to a “ceiling effect,” the baseline value being high. Furthermore, previous studies have found that the ADL index (a composite score of Timed Up and Go test, Z-score, walking speed, grip strength and SMI) was not improved when diet was supplemented with 20–30 g/day whey protein for 26 weeks or 2 years [23,32]. It was also reported that muscular strength was increased when diet was supplemented with whey protein at 20 g/day for 12 weeks [5], 1.7–1.8 g/kgw/day for 16 weeks [22], 21 g/day for 26 weeks, or 30 g/day for 2 years [32], although no increase in muscle mass was detected [5,22,23,32].

A meta-analysis suggested that increasing protein intake to 1.3 g/kgw/day can increase lean body mass dose-dependently in participants without cancer, infectious diseases, or chronic kidney disease and a mean age of 47.2 years [33]. However, In the present study, although mean protein intake in the IC group was increased from 1.0 to 1.3 g/kgw/day, a dose-dependent relationship between protein intake and muscle mass could not be established. The reason for this discrepancy may be that our participants were aged 75 years and older and had multiple complications including cancer, fractures, and pneumonia. Thus, for elderly patients, in addition to increase total daily dietary protein intake and adjusting the protein intake ratios among the three meals of the day, additional resistance exercise may be required to increase muscle mass [34]. Encouragingly, the increase in protein intake up to 1.3 g/kg/day did not impact kidney function in the IC group. Previous studies had suggested that high protein intake may contribute to decreased renal function in elderly patients with chronic kidney disease stage G3 or G4 [35], therefore recommending limiting protein intake to 0.8 g/kg ideal body weight in such patients [36]. On the other hand, it has been shown recently that high protein intake is not associated with renal dysfunction even in people over 90 years of age if renal function is not already impaired [35]. Indeed, another study investigating the relationship between protein intake and renal function suggests that a diet high in protein sources such as soy products may act to protect renal function [37,38]. The present study establishes in older adults that protein intervention consisting primarily of soy products, lean meats and eggs can improve muscle strength without impairing renal function. Meat, especially lean meat as well as chicken, contains a large amount of leucine, which has been found to contribute to improvement of postprandial muscle protein synthesis rate in older adults [4]. Thus, the increased protein intake from soy products, meat and eggs might well underlie the maintenance of renal function despite the high protein intake in this study.

The present study found no significant changes in the ADL-related indices (Barthel index and Lawton score) in either group. Most participants showed favorable ADL by these indexes at baseline, so it was difficult to demonstrate improvement. However, RFD8.75/w, a force development parameter reflecting fundamental motor competence in daily living that is correlated with Timed Up and Go results and routine physical activity and is used to evaluate leg function in older adults with physical frailty [16] tended to be increased in the IC group [16].

This study has several limitations: (1) The nutrient intake at baseline differed between the two groups. Participants were randomly allocated to the groups based on the primary disease for hospitalization, gender, and SMI without regard for dietary energy and macronutrient intake, suggesting the possibility that differences in nutrient intake at baseline may have influenced the end points. (2) A considerable number of participants were excluded from the analysis because they requested a transfer to a general practitioner near their home, but dropout rates were comparable in both groups.

## 5. Conclusions

In adults aged 75 or older with treatable cancer, pneumonia, fractures, and/or urinary-tract infection and at nutritional risk, increased dietary protein intake for 6 months was effective in improving grip strength in comparison with standard care. In addition, the increase in protein intake to 1.3 g/kg/day was found not to impact kidney function.

## Figures and Tables

**Figure 1 nutrients-15-02024-f001:**
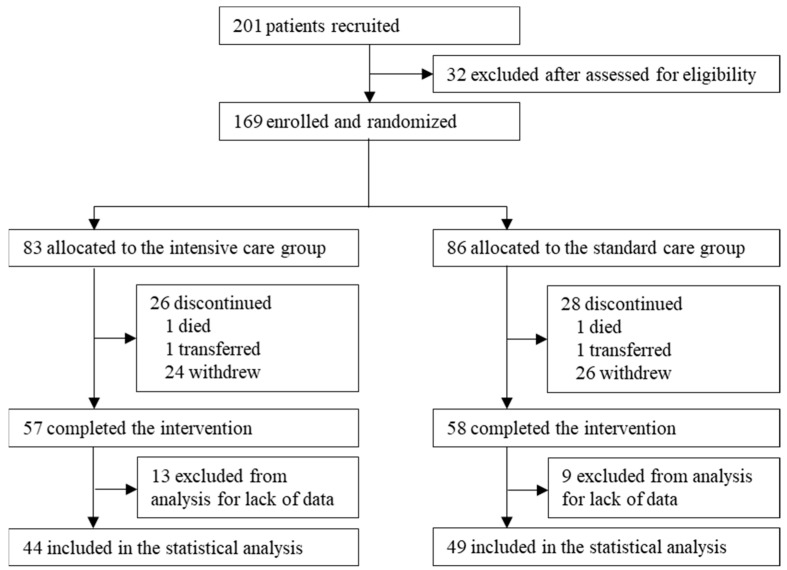
Flow diagram of the study.

**Figure 2 nutrients-15-02024-f002:**
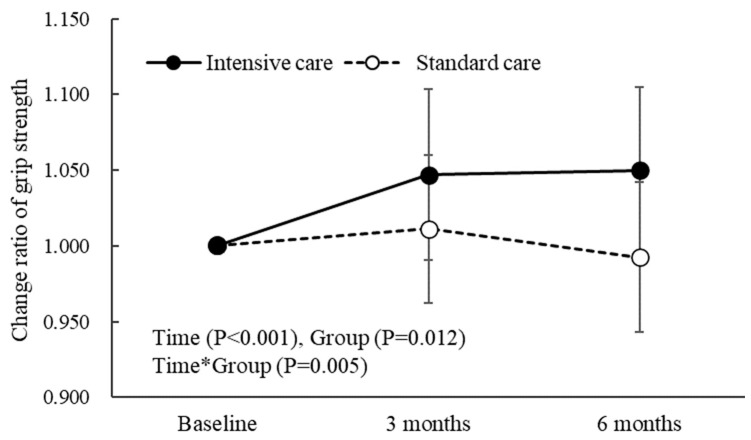
Effects of intervention on grip strength over 6 months. Data are presented as mean ± standard error.

**Table 1 nutrients-15-02024-t001:** Baseline characteristics of the per-protocol population.

	Intensive Care (*n* = 44)	Standard Care (*n* = 49)
Age (years)	79.0 (78.0–84.0)	79.0 (77.0–84.0)
Male	23 (52.3)	28 (57.1)
Female	21 (47.7)	21 (42.9)
BMI (kg/m^2^)	22.3 (21.1–24.7)	21.3 (19.8–24.2)
Grip strength (kg)	20.0 (15.0–28.2)	22.7 (17.1–29.2)
SMI (kg/m^2^)	6.2 (5.8–7.4)	6.4 (5.8–7.0)
F/w (kgf·kg^−1^)	1.19 (1.14–1.27)	1.23 (1.14–1.29)
RFD8.75/w (kgf/s·kg^−1^)	7.79 (5.76–9.69)	7.47 (5.23–9.44)
Barthel index (points)	100.0 (100.0–100.0)	100.0 (100.0–100.0)
Lawton score (points)	8.0 (5.8–8.0)	7.0 (5.0–8.0)
Hb (g/dL)	11.9 (11.0–13.7)	12.0 (11.2–13.2)
Cre (mg/dL)	0.76 (0.62–0.96)	0.77 (0.65–0.97)
BUN (mg/dL)	14.0 (11.0–18.0)	15.0 (12.0–17.0)
eGFR (ml/min/1.73 m^2^)	65.9 (55.6–73.0)	61.8 (53.5–73.8)
Albumin (g/dL)	3.5 (3.3–4.0)	3.5 (3.1–3.9)
CRP (mg/dL)	0.5 (0.2–1.5)	0.6 (0.3–2.0)
Energy intake (kcal/day)	1473.0 (1278.3–1621.8)	1557.7 (1418.5–1705.5) *
Energy intake per body weight (kcal/kgw/day)	24.4 (22.5–28.4)	29.1 (24.7–33.2) *
Protein intake (g/day)	59.0 (52.2–65.2)	62.1 (56.3–70.1)
Protein intake per bodyweight (g/kgw/day)	1.0 (0.9–1.2)	1.2 (1.0–1.4) *
Fat intake (g/day)	40.7 (33.2–45.6)	42.4 (38.2–49.1)
Carbohydrate intake (g/day)	211.7 (176.1–236.8)	222.6 (195.3–245.1) *
Vitamin D (μg/day)	6.2 (4.4–7.7)	5.8 (4.1–8.1)
Physical activity (kcal/day)	108.4 (35.8–196.2)	39.1 (0.0–228.2)
Primary disease on admission		
Cancer	23 (52.3)	28 (57.1)
Fractures	8 (18.2)	7 (14.3)
Urinary-tract infection	1 (2.3)	1 (2.0)
Pneumonia	12 (27.3)	13 (26.5)
Concomitant disease (%)		
Diabetes	14 (31.8)	21 (42.9)
Hypertension	23 (52.3)	34 (69.4)
Dyslipidemia	15 (34.1)	20 (40.8)
Osteoporosis	0 (0.0)	0 (0.0)
Cancer	4 (9.1)	5 (10.2)
Chronic kidney disease	2 (4.5)	3 (6.1)
Liver dysfunction	2 (4.5)	0 (0.0)

Data are presented as median (interquartile range) for quantitative data and as *n* (%) for categorical data. * *p* < 0.05, analyzed by *t*-test or Mann–Whitney U test for quantitative data and Fisher’s exact test for categorical data. The concomitant diseases are those of the participants in addition to their primary disease at admission. BMI: body mass index; SMI: skeletal muscle index; F/w: peak reaction force per unit body weight; RFD8.75/w: maximal rate of force development (Δ87.5 ms) per unit body weight; Hb: hemoglobin; Cre: creatinine; BUN: blood urea nitrogen; eGFR: estimated glomerular filtration rate; CRP: C-reactive protein.

**Table 2 nutrients-15-02024-t002:** Summary of post-intervention effect on outcomes.

	Intensive Care	Standard Care	Between-Group
Grip strength (kg)	1.11 (0.11 to 2.10) *	−0.18 (−1.08 to 0.71)	1.29 (0.22 to 2.36) *
SMI (kg/m^2^)	0.07 (0.02 to 0.11) **	0.04 (0.01 to 0.07) **	0.02 (−0.05 to 0.09)
Barthel index (points)	1.71 (−0.15 to 3.56)	0.61 (−0.51 to 1.74)	1.16 (−0.99 to 3.30)
Lawton score (points)	0.30 (−0.17 to 0.76)	0.10 (−0.46 to 0.66)	0.41 (−0.19 to 1.02)
F/w (kgf·kg^−1^)	0.25 (−0.08 to 0.58)	0.02 (−0.27 to 0.31)	0.01 (−0.04 to 0.04)
RFD8.75/w (kgf/s·kg^−1^)	0.90 (0.27 to 1.53) **	0.76 (0.01 to 1.51) *	0.23 (−0.72 to 1.19)
BMI (kg/m^2^)	0.79 (0.30 to 1.27) **	0.36 (−0.25 to 0.75)	0.73 (−0.46 to 1.93)
Hb (g/dL)	0.31 (−0.28 to 0.90)	0.80 (0.35 to 1.24) **	0.05 (−0.60 to 0.70)
Cre (mg/dL)	0.05 (−0.02 to 0.11)	0.02 (−0.03 to 0.08)	0.06 (−0.05 to 0.16)
BUN (mg/dL)	2.28 (0.48 to 4.08) **	2.20 (0.37 to 4.04) *	0.92 (−0.93 to 2.77)
eGFR (ml/min/1.73 m^2^)	−5.40 (−10.94 to 0.15)	−3.26 (−7.93 to 1.41)	−2.04 (−8.70 to 4.61)
Albumin (g/dL)	0.45 (0.27 to 0.63) **	0.45 (0.27 to 0.63) **	0.03 (−0.12 to 0.17)
CRP (mg/dL)	−0.94 (−1.49 to −0.39) **	−1.20 (−2.09 to −0.32) **	−0.33 (−0.66 to −0.01) ^#^

Data are presented as mean ± 95% confidence interval. ** *p* < 0.01, * *p* < 0.05 (vs. Baseline), ^#^ *p* < 0.05 analyzed by ANOVA. SMI: skeletal muscle index; F/w: peak reaction force per unit body weight; RFD8.75/w: maximal rate of force development (Δ87.5 ms) per unit body weight; BMI: body mass index; Hb: hemoglobin; Cre: creatinine; BUN: blood urea nitrogen; eGFR: estimated glomerular filtration rate; CRP: C-reactive protein.

**Table 3 nutrients-15-02024-t003:** Post-intervention effects on intake of food groups that are protein sources.

	Intensive Care		Standard Care	
	Baseline	6 Months	Baseline	6 Months
Soy products (g)	31.4 (25.1–42.3)	70.0 (35.0–90.0) **	38.9 (21.3–51.8)	45.0 (25.0–70.0)
Seafood (g)	63.0 (47.7–82.0)	73.6 (45.7–92.1)	58.3 (38.2–89.3)	71.4 (50.0–100.0)
Meat (g)	55.6 (40.5–76.9)	74.3 (50.0–98.6) **	55.6 (37.5–84.0)	68.6 (45.7–91.4)
Eggs (g)	27.4 (18.4–37.9)	46.4 (24.1–50.0) *	33.3 (23.3–42.1)	50.0 (21.4–50.0)
Dairy products (g)	118.7 (69.4–148.6)	149.5 (72.9–232.5)	98.9 (60.2–122.9)	159.6 (62.5–232.5)

Data are presented as median (interquartile range). ** *p* < 0.01, * *p* < 0.05 (vs. Baseline) analyzed by ANOVA.

## Data Availability

The datasets generated during and/or analyzed during the current study are available from the corresponding author on reasonable request.

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
