# Peer review of "Efficacy and Safety of 6-Month High Dietary Protein Intake in Hospitalized Adults Aged 75 or Older at Nutritional Risk: An Exploratory, Randomized, Controlled Study"

_nutrients, 2023, doi:10.3390/nu15092024_

Round 1

Reviewer 1 Report

I consider this article to be very relevant on the subject, above all a contribution to increasing knowledge on the relationship between protein consumption in patients at nutritional risk with various pathologies, namely treatable cancer and its relationship with the various parameters evaluated in two baseline moments and after the intervention, also warns of the need for more research in this area of knowledge.

 I therefore make some suggestions that I believe may be a contribution to clarifying less clear points:

 - I suggest that in the title and throughout the text instead of ……..mild malnutrition should be at nutritional risk;

 -I suggest that in the abstract in paragraph 44 ….it seems to me that instead of 1.5/kgweight/day it should be 1.3g/kgweight/day?;

In table 3: What vegetables are soy products? It is not clear!

 Materials and methods

Line 80……. Replace mild malnutrition with nutritional risk;

Participants

Line 91 ………In the mini nutritional assessment short form: between 8 and 11 is risk of malnutrition and not mild malnutrition Has the mini nutritional assessment short form been validated in Japan? The reference should be.

Procedures

Line 137……….. (FFQg) Was it validated and authorized by the authors? has no reference

 Line 141……specific methods……..Only with foods but not only foods rich in VBA proteins why did you consider soy?

 Line 143………..Did they not even provide information on the duration of the exercises?

 Line 145…….. What are the cutoff points for men and women; what methodology is used? How many measurements? Which hand used left or right?

 Line 152…….. What are the cutoff points for the bioimpedance assessment? And what are the prerequisites for bioimpedance assessment? They should be described.

DISCUSSION

Line 245 …….who also have a mild nutritional deficiency replace with nutritional risk.

Line 292……Limitation 2, I do not consider it to be a limitation but rather a conclusion: because it means that in the studied groups that did not have kidney disease, this increase in protein intake up to 1.3g/kg/day was not harmful to the kidney .

In the discussion, I would have liked to have addressed the influence of pathologies on the results The high consumption of fats to the type of proteins ingested; to the average time of practice of physical exercise.

The conclusions

They are insufficient in relation to the research work, for example they do not refer to the fact that a high-protein diet of up to 1.3g/kg/day did not cause injury in these participants. The importance of not selecting patients with kidney disease when intending to intervene with hyperprotein nutritional plans….

Author Response

Comment 1-1:

I consider this article to be very relevant on the subject, above all a contribution to increasing knowledge on the relationship between protein consumption in patients at nutritional risk with various pathologies, namely treatable cancer and its relationship with the various parameters evaluated in two baseline moments and after the intervention, also warns of the need for more research in this area of knowledge. I therefore make some suggestions that I believe may be a contribution to clarifying less clear points:

Response 1-1:

Thank you very much for your valuable comments and suggestions on our manuscript, which we have revised accordingly.

Comment 1-2:

I suggest that in the title and throughout the text instead of ……..mild malnutrition should be at nutritional risk;

Response 1-2:

Thank you very much for your suggestion. We have revised the manuscript accordingly.

Comment 1-3:

I suggest that in the abstract in paragraph 44 ….it seems to me that instead of 1.5/kg weight/day it should be 1.3g/kg weight/day?

Response 1-3:

Thank you for your question. We had asked the patients in the intervention group to increase their protein intake to 1.5 g/kgw/day, but it was increased only to 1.3. We revised the sentence as follows:

Line 45, “…the protein intake goals (g/kgw/day) were 1.5 for IC and 1.0 for SC”.

Comment 1-4:

In table 3: What vegetables are soy products? It is not clear!

Response 1-4:

Thank you for your question. We changed “Legumes” to “Soy products” in Table 3.

Comment 1-5:

Line 80……. Replace mild malnutrition with nutritional risk;

Response 1-5:

Thank you very much for your suggestion. We have made the change thought the manuscript.

Comment 1-6:

Line 91 ………In the mini nutritional assessment short form: between 8 and 11 is risk of malnutrition and not mild malnutrition Has the mini nutritional assessment short form been validated in Japan? The reference should be.

Response 1-6:

Thank you very much for your question. While the mini nutritional assessment short form has not been validated in Japan, it has been used previously. We made the following change and provided a reference:

Lines 92-93, “…scoring below 11 on the Mini Nutritional Assessment Short Form, a simple evaluation questionnaire for nutritional state [12, 13]”

Reference.

  1. Rubenstein LZ, Harker JO, Salvà A, Guigoz Y, Vellas B: Screening for undernutrition in geriatric practice: developing the short-form mini-nutritional assessment (MNA-SF). J Gerontol A Biol Sci Med Sci 2001, 56:M366-372.
  2. Izawa S, Enoki H, Hasegawa J, Hirose T, Kuzuya M. J Nutr Health Aging. 2014;18(4):372-7.

Comment 1-7:

Line 137. (FFQg) Was it validated and authorized by the authors? has no reference.

Response 1-7:

Thank you for your question. We added reference 16 on validation of FFQg.

Reference.

  1. Takahashi K, Yoshimura Y, Kaimoto T, Kunii D, Komatsu T, Yamamoto S. Validation of a food frequency questionnaire based on food groups for estimating individual nutrient intake. Jpn J Nutr 2001, 59: 221-232.

Comment 1-8:

 Line 141……specific methods……..Only with foods but not only foods rich in VBA proteins why did you consider soy?

Response 1-8:

Thank you for your question. The study was designed to investigate increased protein intake in ordinary meals not involving protein-enriched supplements.

Comment 1-9:

 Line 143………..Did they not even provide information on the duration of the exercises?

Response 1-9:

Thank you very much for your question. We have included more information on the instructions for light resistance exercise as follows:

Line 136-141, “Neither group received individualized advice on exercise; both groups received a leaflet asking the participants to perform light resistance exercise 6 times a week (3 exercises, each one repeated 10 times but not on two consecutive days) for the arms and legs (side lateral raise and thigh lift exercise).”

Comment 1-10:

 Line 145…….. What are the cutoff points for men and women; what methodology is used? How many measurements? Which hand used left or right?

Response 1-10:

Thank you very much for your question. We included more information on grip strength measurement. In addition, references were added for cutoff values of grip strength and SMI.

Lines 147-150, “To measure grip strength, the subject held a hand dynamometer downward in the standing position; grip strength was measured twice on each side at maximum effort while exhaling and the maximum value was used as upper limb muscle strength.”

Lines 154-157, “The cutoff point of grip strength and SMI were determined by the diagnostic criteria of the Asian Working Group for Sarcopenia (AWGS2019) (SMI: < 7.0 kg/m2 for male and < 5.7 kg/m2 for female; grip strength: < 28.0 kg for male and < 18.0 kg for female) [21].”

Comment 1-11:

Line 152…….. What are the cutoff points for the bioimpedance assessment? And what are the prerequisites for bioimpedance assessment? They should be described.

Response 1-11:

Thank you for your suggestion. We have included the cutoff value of SMI in lines 164-166. We included the prerequisites for bioimpedance assessment as follows:

Lines 151-154, “SMI was evaluated by bioelectrical impedance analysis using a multi-frequency body composition meter at seated position after 5 minutes resting in accordance with the general instructions by the manufacturer (Model JMW140, InBody S10; Biospace Co., Ltd., South Korea) [20].” 

Comment 1-12:

Line 245 …….who also have a mild nutritional deficiency replace with nutritional risk.

Response 1-12:

We have revised the manuscript accordingly.

Comment 1-13:

Line 292……Limitation 2, I do not consider it to be a limitation but rather a conclusion: because it means that in the studied groups that did not have kidney disease, this increase in protein intake up to 1.3g/kg/day was not harmful to the kidney.

Response 1-13:

Thank you for your valuable suggestion. We moved limitation 2 to Conclusion as follows:

Lines 323-324, “…the increase in protein intake to 1.3 g/kg/day did not impact kidney function in the IC group.”

Comments 1-14:

In the discussion, I would have liked to have addressed the influence of pathologies on the results The high consumption of fats to the type of proteins ingested; to the average time of practice of physical exercise.

Response 1-14:

Thank you for your comment. We now mention the effect of physical activity and type of protein intake on muscle strength and muscle mass improvement in Discussion.

Lines 264-267, “According to IPAQ quantification, the physical activity recommended to both groups was light-intensity activity such as walking 3 to 5 times per week; since physical activity of the two groups was similar, exercise should not be a factor in the observed increase of muscle strength and mass in the IC group.”

Lines 295-304, “…another study investigating the relationship between protein intake and renal function suggests that a diet high in protein sources such as soy products may act to protect renal function [39,40]. The present study establishes in older adults that protein intervention consisting primarily of soy products, lean meats and eggs can improve muscle strength without impairing renal function. Meat, especially lean meat as well as chicken, contain a large amount of leucine, which has been found to contribute to improvement of the postprandial muscle protein synthesis rate in older adults [4]. Thus, the increased protein intake in the IC group in our study being substantially from increased intake of soy products, meat and eggs might well underlie their maintenance of renal function despite the high protein intake.”

Comments 1-15:

They are insufficient in relation to the research work, for example they do not refer to the fact that a high-protein diet of up to 1.3g/kg/day did not cause injury in these participants. The importance of not selecting patients with kidney disease when intending to intervene with hyperprotein nutritional plans….

Response 1-15:

Thank you very much for your constructive comments. We have now included discussion of the safety and efficacy of protein-enriched intervention as follows:

Lines 289-304, “Encouragingly, the increase in protein intake up to 1.3 g/kg/day did not impact kidney function in the IC group. Previous studies had suggested that high protein intake may contribute to decreased renal function in elderly patients with chronic kidney disease stage G3 or G4 [37], therefore recommending limiting protein intake to 0.8 g/kg ideal body weight in such patients [38]. On the other hand, it has been shown recently that high protein intake is not associated with renal dysfunction even in people over 90 years of age if renal function is not already impaired [37]. Indeed, another study investigating the relationship between protein intake and renal function suggests that a diet high in protein sources such as soy products may act to protect renal function [39,40]. The present study establishes in older adults that protein intervention consisting primarily of soy products, lean meats and eggs can improve muscle strength without impairing renal function. Meat, especially lean meat as well as chicken, contains a large amount of leucine, which has been found to contribute to improvement of postprandial muscle protein synthesis rate in older adults [4]. Thus, the increased protein intake in the IC group in our study being substantially from increased intake of soy products, meat and eggs might well underlie the maintenance of renal function despite the high protein intake.”

Reviewer 2 Report

Efficacy and safety of 6 months intensive dietary protein intervention in hospitalized older adults with mild malnutrition.

 On what basis did authors checked the safety of the  nutritional  intervention?  I would  recommend to refer to grip  strength. And was the intervention intensive? Participants got the  dietetic recommendations,  not hot specific high protein diet. The outcome may be differenced.

In the abstract, the author should consider whether to use the term “subjects” and “older adults”.

-2.3 Target sample size and rationale. This whole paragraph needs to be reformulated to understand it clearly.  Line 115. Minimum number or participants was set at 200 or 169? Or do authors mean, initially 200 was selected? It is rather confusing because in figure 1 it is shown that 201 patients are assessed for eligibility. Clarify to make it sound understandable.

-outcomes- line 167
The word changes is best replaced with (changed) instead.

-In figure 1

 169 enrolled and randomized
26 participants and 28 participants discontinued respectively. However, it is confusing whether the rest of the consequent numbers are sub-values or utterly separate.

 Table 1. Have the authors performed the comparison  of intervention and Standard care? Some differences seem to be significant, as i.e.  physical activity.

What is the difference between cancer and malignancy?

Why data are presented as mean and standard deviation? All data met the criteria of homogeneity of variance and normal distribution?( that’s refers not only to the  table 1 but whole study)

 Protein intake? Is there no difference at the baseline? After the intervention the intervention group reached the level of  control group of protein intake? Have the Authors considered what might be the cause of  that initial difference?

-Line 271
A meta-analysis of studies of muscle mass “by” increased protein dietary intervention. It is better to replace by to “with”.

-Procedures- Line 130 ,131 and 145
Repetitively use of the word “and”. Sentence could be simplified to make it simpler to read.

Some of the sentences are quite difficult to digest. For example: line 193-194. 201-203. Etc.

In the result section, line 192.
 Authors may have left the word “out” before of the 169 subjects.

Generally, English use is to be revised. There is abundance of repetitions like line 94 254

Author Response

Comment 2-1:

On what basis did authors checked the safety of the nutritional intervention? I would recommend to refer to grip strength. And was the intervention intensive? Participants got the dietetic recommendations, not hot specific high protein diet. The outcome may be differenced.

Response 2-1:

Thank you very much for your critical review and constructive comments on our manuscript. First, our assessment of the safety of the nutritional intervention includes blood tests (e.g., renal and hepatic function) and physical examinations before and after initiation of the intervention. Second, we designed this study to evaluate the usefulness of protein fortification in daily-life settings and therefore did not use protein-enriched supplements.

Lines 140-141, “All outcomes were assessed at baseline and 3- and 6-months after intervention and included blood tests (renal and hepatic function).”

Comment 2-2:

In the abstract, the author should consider whether to use the term “subjects” and “older adults”.
Response 2-2:

Thank you very much. We now use “study participants” instead of “subjects” and “adults aged 75 or older” instead of “older adults” throughout the manuscript.

Comment 2-3:

2.3 Target sample size and rationale. This whole paragraph needs to be reformulated to understand it clearly. Line 115. Minimum number or participants was set at 200 or 169? Or do authors mean, initially 200 was selected? It is rather confusing because in figure 1 it is shown that 201 patients are assessed for eligibility. Clarify to make it sound understandable.

Response 2-3:

Thank you very much for your remark. We have clarified the sample size and selection process as follows:

Lines 110-112, “As this was an exploratory randomized controlled trial, 200 participants were set as the target sample size to be recruited through the participating facilities during the study period. One hundred sixty-nine patients were finally enrolled and assigned to the study.”

Comment 2-4:

outcomes- line 167. The word changes is best replaced with (changed) instead.

Response 2-4:

Thank you very much for your suggestion. We revised the manuscript as follows:

Lines 169-170, “The secondary endpoint was change in macronutrient intake over 6 months.”

Comment 2-5:

In figure 1. 169 enrolled and randomized 26 participants and 28 participants discontinued respectively. However, it is confusing whether the rest of the consequent numbers are sub-values or utterly separate.

Response 2-5:

Thank you very much for your clarification. We revised Figure 1.

Comment 2-6:

Table 1. Have the authors performed the comparison of intervention and standard care? Some differences seem to be significant, as i.e. physical activity. What is the difference between cancer and malignancy?

Response 2-6:

Table 1 compares the baseline characteristics of participants in the IC and SC groups. Statistically significant differences were noted in energy intake (kcal/day), energy intake (kcal/kwg), protein intake (g/kgw) and carbohydrate intake (g/day), while no statistically significant difference was noted in many items including physical activity. We used “cancer” in primary disease at admission and “malignancy” in comorbidities when participants had cancers in addition to the primary disease of admission. We now use “cancer” in both cases.

Comment 2-7:

Why data are presented as mean and standard deviation? All data met the criteria of homogeneity of variance and normal distribution? (that’s refers not only to the table 1 but whole study)

Response 2-7:

Thank you very much for your constructive comment. We have now performed the Kolmogorov-Smirnov normality test to test the normality of the data and revised the manuscript accordingly.

Comment 2-8:

Protein intake? Is there no difference at the baseline? After the intervention the intervention group reached the level of control group of protein intake? Have the Authors considered what might be the cause of that initial difference?

Response 2-8:

Thank you very much for your questions. As you pointed out, protein intake per body weight (g/kgw/day) but not protein intake (g/day) at baseline significantly differs between the two groups, as shown in Table 1. We believe that this is partly due to protein intake not being taken into consideration for randomization, the participants in IC group having slightly higher BMI. Protein intake per bodyweight (g/kgw/day) was significantly improved after 6 months in the IC group but not in the SC group, making protein intake per bodyweight (g/kgw/day) comparable between the two groups although protein intake (g/day) was higher in the IC group. We have included as a limitation of the study that protein intake at baseline was not taken into consideration at randomization.

Comment 2-9:

Line 271. A meta-analysis of studies of muscle mass “by” increased protein dietary intervention. It is better to replace by to “with”.

Response 2-9:

Thank you very much for your suggestion. We revised the sentence.

Comment 2-10:

Procedures- Line 130 ,131 and 145. Repetitively use of the word “and”. Sentence could be simplified to make it simpler to read. Some of the sentences are quite difficult to digest. For example: line 193-194. 201-203. Etc.
Response 2-10:

Thank you very much for your remarks. We made revisions accordingly.

Comment 2-11:

In the result section, line 192. Authors may have left the word “out” before of the 169 subjects.
Response 2-11:

Thank you very much for your suggestion. We added the word “out” before “of the 169 subjects”.

Comment 2-12:

Generally, English use is to be revised. There is abundance of repetitions like line 94 254
Response 2-12:

We have asked our native language editor to polish the manuscript.

Round 2

Reviewer 2 Report

The composition in terms of writings and literature are somehow difficult to understand when reading at first time. In a way, more than once time reading is needed to comprehend on what is being written. Besides, there’s some error that could be avoided and precise of English use.

Efficacy and safety of 6-month increased dietary protein intervention in hospitalized adults aged 75 or older at nutritional risk: An exploratory, randomized, controlled study.

For example, the title. Title could be direct and understandable. When using the word intervention, one could not increase intervention but rather changed, improve, effective and etc.

Line 67-69 and line 162-164

Previous studies suggested that 12-week protein-enrichment intervention was effective in improving physical function in older adults but medium- to long-term effects remained unclear due to the relatively short duration of the interventions.

which includes questions regarding the number of days per week on which the subject performed mod- erate- to high-intensity physical activities

-Does so many dash/hyphen needed?

Line 125

Spelling correction – judgement

Line 71

Spelling correction- Sparse

Next, line 135 (intake_but).

Line 278- 281

“A meta-analysis of studies of muscle mass by increased protein dietary intervention  in participants without cancer, infectious diseases, or chronic kidney disease with a mean  age of 47.2 years (range: 19-81 years) suggested that even without resistance exercises,  increased protein intake to 1.3 g/kgw/day can contribute to a dose-dependent increase in  lean body mass.”

-This sentence seems to be overly lengthy and difficult for readers to understand.

 Line 286-287

-to increase will be better than “to increasingly”

Line 302-304

“Thus, the increased protein intake in the IC group in our study being substantially from increased intake of soy products, meat and eggs might well underlie the maintenance of renal function despite the high protein intake.”

-Sentence could be modified.

Line 300.

-Besides, the whole manuscript should have a standardized fonts Pay attention to line 300.

Recommendation: to be check thoroughly by an English native speaker.

Author Response

Reviewer 2

Comment 2-1:

The composition in terms of writings and literature are somehow difficult to understand when reading at first time. In a way, more than once time reading is needed to comprehend on what is being written. Besides, there’s some error that could be avoided and precise of English use.

Efficacy and safety of 6-month increased dietary protein intervention in hospitalized adults aged 75 or older at nutritional risk: An exploratory, randomized, controlled study.

For example, the title. Title could be direct and understandable. When using the word intervention, one could not increase intervention but rather changed, improve, effective and etc.

Response 2-1:

Thank you very much for your instructive comments. We have revised our manuscript based on your valuable inputs including the title.

New title, “Efficacy and safety of 6-month high dietary protein intake in hospitalized adults aged 75 or older at nutritional risk: An exploratory, randomized, controlled study.”

Comment 2-2:

Line 67-69 and line 162-164

Previous studies suggested that 12-week protein-enrichment intervention was effective in improving physical function in older adults but medium- to long-term effects remained unclear due to the relatively short duration of the interventions.

which includes questions regarding the number of days per week on which the subject performed mod- erate- to high-intensity physical activities

-Does so many dash/hyphen needed?

Response 2-2:

Thank you very much for your clarification. We have revised the manuscript accordingly.

Comment 2-3:

Line 125

Spelling correction – judgement

Line 71

Spelling correction- Sparse

Next, line 135 (intake_but).

Response 2-3:

Thank you very much for your inputs. We have revised the manuscript accordingly.

Comment 2-4:

Line 278- 281

“A meta-analysis of studies of muscle mass by increased protein dietary intervention in participants without cancer, infectious diseases, or chronic kidney disease with a mean age of 47.2 years (range: 19-81 years) suggested that even without resistance exercises, increased protein intake to 1.3 g/kgw/day can contribute to a dose-dependent increase in lean body mass.”

-This sentence seems to be overly lengthy and difficult for readers to understand.

Response 2-4:

Thank you very much for your inputs. We have revised the manuscript as below:

Lines 278-280, “A meta-analysis suggested that increasing protein intake to 1.3 g/kgw/day can increase lean body mass dose-dependently in participants without cancer, infectious diseases, or chronic kidney disease and a mean age of 47.2 years [35].”

Comment 2-5:

 Line 286-287

-to increase will be better than “to increasingly”

Response 2-5:

Thank you very much for your clarification. We have revised the manuscript accordingly.

Comment 2-6:

Line 302-304

“Thus, the increased protein intake in the IC group in our study being substantially from increased intake of soy products, meat and eggs might well underlie the maintenance of renal function despite the high protein intake.”

-Sentence could be modified.

Response 2-6:

Lines 300-302, “Thus, the increased protein intake from soy products, meat and eggs might well underlie the maintenance of renal function despite the high protein intake in this study.”

Comment 2-7:

Line 300.

-Besides, the whole manuscript should have a standardized fonts Pay attention to line 300.

Response 2-7:

Thank you very much for your clarification. We have revised the manuscript accordingly.